# Bootstrapping World Models from Dynamics Models in Multimodal Foundation Models

## Abstract

To what extent do vision-and-language foundation models possess a realistic world model (observation × action → observation) and a dynamics model (observation × observation → action), when actions are expressed through language? While open-source foundation models struggle with both, we find that fine-tuning them to acquire a dynamics model through supervision is significantly easier than acquiring a world model. In turn, dynamics models can be used to bootstrap world models through two main strategies: 1) weakly supervised learning from synthetic data and 2) inference time verification. Firstly, the dynamics model can annotate actions for unlabelled pairs of video frame observations to expand the training data. We further propose a loss-weighting mechanism for the image tokens weighted by the its importance predicted by a recognition model. Secondly, the dynamics models can assign rewards to multiple samples of the world model to score them, effectively guiding search at inference time. We evaluate the world models resulting from both strategies through the task of *action-centric image editing* on Aurora-Bench. Our best model achieves a performance competitive with state-of-the-art image editing models, improving on them by a margin of 15% on real-world subsets according to GPT4o-as-judge, and achieving the best average human evaluation across all subsets of Aurora-Bench.[1].

## 1 Introduction

*World models* (observation × action → observation) (Ha and Schmidhuber, 2018; Agarwal et al., 2025; Bruce et al., 2024; Brooks et al., 2024) can be successfully trained to simulate future trajectories given the history of past observations and actions. World models are instrumental in training embodied agents to endow them with specific abilities (Qin et al., 2024), such as grounding on affordances (Brohan et al., 2023), spatio-temporal reasoning (Huang et al., 2022; Li et al., 2025), and planning (Reed et al., 2022a; Yang et al., 2023; Hafner et al., 2025). However, learning a specialised world model is challenging. Firstly, it requires a large amount of real-world data (Liu et al., 2024) and even this data volume may be insufficient within the confines of the current training paradigm (Motamed et al., 2025). Secondly, the benefit of creating a separate world model to train a downstream embodied agent remains unclear because of possible compounding errors between the two models. Conversely, foundation models, such as vision-language models (VLMs), are already imbued with plenty of real-world knowledge of both action (in language form) and perception (in vision form), because of their large-scale pre-training. While such knowledge is not straightforward to elicit (Gao et al., 2024; Qiu et al., 2024; Abdou et al., 2021), we propose investigating a promising alternative to specialised world models, by enhancing the knowledge implicitly stored inside foundation models.

Firstly, we probe whether native VLMs already contain reliable world models, facilitated by model designs that combine various modalities into a unified representation, i.e., sequences of tokens (Chameleon Team, 2024). In particular, we frame the assessment of world models as the ability to solve *action-centric image editing tasks* (Krojer et al., 2024). In such tasks, the model predicts the next observation given the previous observation and an action expressed as a language instruction. Based on our evaluation, we empirically demonstrate that existing open-source models do not prefer ground-truth trajectories compared to adversarially generated ones. Hence, we verify that the world

---

[1]The code and models used in this paper will be available at `[anonymised]`.

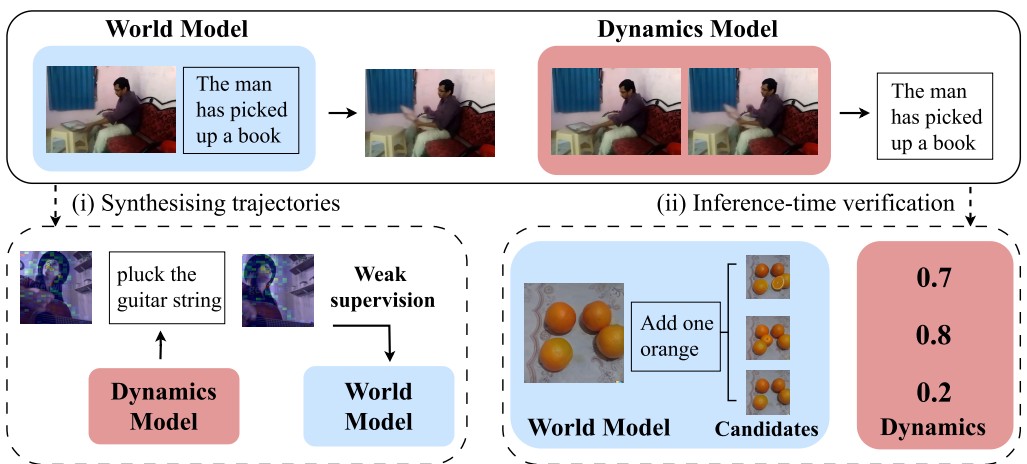

Figure 1: A high-level illustration of our two strategies to bootstrap a world model from a dynamics model in Vision–Language Models: (i) synthesising trajectories for weak supervision (**left**) and (ii) inference-time verification of candidate observations (**right**).

model implicit in the original VLMs *per se* is not well grounded on real-world trajectories (Gao et al., 2024; Qiu et al., 2024; Abdou et al., 2021).

Surprisingly, we also find that acquiring a dynamics model (observation × observation → action) via supervised fine-tuning is substantially easier than directly acquiring a world model (observation × action → observation). Inspired by this observation, we propose two strategies to bootstrap the world model from the dynamics model in a given VLM, namely (i) **learning from synthetic trajectories** in videos automatically labelled with actions by the dynamics model; and (ii) **test-time verification** of predicted observations sampled from the world model through the dynamics model.

For the weak supervision strategy, which is reminiscent of (Baker et al., 2022), we use a dynamics model fine-tuned on the AURORA dataset Krojer et al. (2024) to annotate motion key-frames pairs extracted from real-world videos with actions (in language form). Around 45 hours of unlabelled videos are sourced from movements-in-time (Monfort et al., 2019), Kinetics700 (Kay et al., 2017; Carreira et al., 2019) and UCF-101 (Soomro et al., 2012). Together with the ground-truth trajectories in AURORA, the synthesised trajectory triplets (observation × annotated action → observation) are then used for supervised fine-tuning of the VLM world model. To effectively train the world model, we additionally propose a *loss-weighting method* which weights the loss of each image token according to the visual difference between the ground-truth source and target observations, as estimated by a recognition model. In the verification strategy, we show how using the VLM dynamics model to assign rewards to multiple samples generated by the VLM world model can effectively guide search at inference time.

We conduct an extensive evaluation on MagicBrush, Action-Genome, Something-Something, What-sUp and Kubric in AURORA-BENCH (Krojer et al., 2024). We focus on Chameleon-7B as the best available open-source foundation model, and transform it into a world model (**CWM**; **C**hameleon **W**orld **M**odel). We show that thanks to the synthetic data strategy to bootstrap world models from dynamics models, our general-purpose CWM can achieve an overall performance superior to state-of-the-art diffusion models specialised for image editing. In particular, CWM improves GPT4o-as-a-judge scores on the Something-Something, Action-Genome, and Kubric subsets of AURORA by 15%, 15% and 7%, respectively. Similarly, human evaluators rate CMW image editing consistently better. Inference-time verification can also improve AURORA-finetuned Chameleon to a comparable degree as data synthesis, providing an effective training-free bootstrapping method. In some cases, it can even be combined for compounded gains. To summarise our contributions:

- We empirically show that VLMs like Chameleon-7B do not exhibit a clear preference for ground-truth real-world trajectories over heuristic-generated incorrect ones.

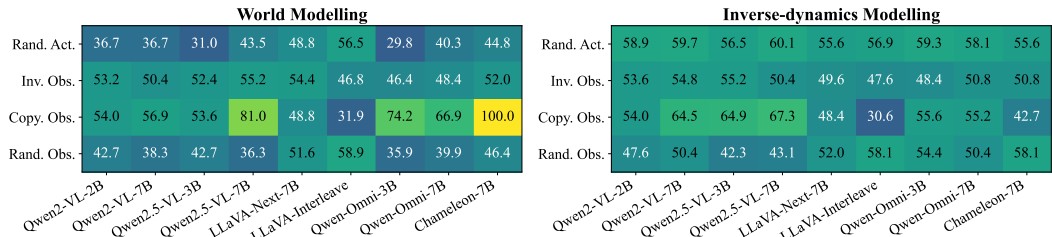

Figure 2: Percentages of 9 VLMs' preferences towards Reference vs. 4 types of Negatives, across world modelling and inverse-dynamics modelling tasks. Higher values are better.

- We propose two strategies to bootstrap a world model from a dynamics model inside VLMs: (i) learning from unlabelled videos annotated with actions by a dynamics model, and (ii) verifying the generated observations with the dynamics model at inference time.
- We conduct extensive evaluations on AURORA-BENCH: both GPT4o-as-a-judge and human raters demonstrate the effectiveness of our methods with a considerable margin compared to the state-of-the-art image editing models.

## 2  VLMs Lack a Consistent Preference for Real-World Trajectories

The first research question we investigate in this paper is: **To what extent do VLMs exhibit a preference for token sequences of actions and observations that align with real-world trajectories?**

To address this question, we evaluate 9 VLMs on ground-truth trajectories from 5 subsets of AURORA-BENCH (Krojer et al., 2024): MagicBrush, Something-Something, Action-Genome, Whatsup, and Kubric. Each subset contains 50 trajectory triplets of the form $(o_s, a, o_t)$, where $o_s$ is the source observation, $a$ the action text, and $o_t$ the next observation. [2]

We then manually curate four types of negative trajectories using rules: two that manipulate the observation of the trajectory triplet, and two that manipulate the action. We design two kinds of action-level manipulation: 1) **Random Action**: for a given pair of observations, we substitute the original action with another randomly sampled within the same subset. 2) **Random Observation**: we randomly substitute the target observation with another in the same subset. We also test the following observation-level manipulations. 3) **Copy Observation**: we directly copy the source observation as the target observation. 4) **Inverse Observation**: we swap the source and target observations.

In Figure 2, we compare the predicted log-likelihood VLMs assign to each ground-truth trajectory (Reference) against its corresponding manipulated one (Negatives). We evaluate the VLMs in two tasks: action prediction (i.e., as a dynamics model) and next-observation prediction (i.e., as a world model). For each kind of negative trajectory, we report the percentage of samples where the model favours the reference trajectory over the negative trajectory. From Table **??**, it emerges that VLMs display a very limited preference for the ground-truth trajectories in a zero-shot setting (around 50%). In the action prediction task (right panel), there is a slightly higher tendency to favour the ground-truth over the group with random actions; however, even in the best case, Qwen2.5-VL-7B prefers the reference in only 60.08% of the samples. The only negative group that seems to be identifiable for VLMs is the inverse observation, where Qwen2.5-VL-7B has 67.34% of correct preference. In the next-observation prediction task (left panel), the VLM mostly fails in effectively differentiating the ground truth from the negatives. An exception to this is the copy manipulation, where the Chameleon can always tell them apart. Although the underlying reason remains uncertain, one plausible explanation for this behaviour is that the model's ability to solve next-observation prediction tasks depends on their alignment with training sequences: for instance, it is plausible that Chameleon's data rarely features two identical adjacent images. We provide a breakdown discussion for Chameleon in Appendix F.

---

[2] We choose these 9 VLMs with the consideration of 1) they are public accessible and 2) we ensure that they have been exposed to interleaved data during their pre-training.

Figure 3: Heatmap visualization of image token weights predicted by the recognition model on examples from UCF-101, Something-Something, MagicBrush, and Kubric.

# 3 BOOTSTRAPPING A WORLD MODEL FROM A DYNAMICS MODEL IN VLMS

Since we showed in Section 2 that Chameleon-7B displays a higher proclivity as for action prediction than next-observation prediction, we first verify that this tendency is intensified when Chameleon-7B is fine-tuned on image editing trajectories (Section 3.1), as this results in the VLM acting reliably as a dynamics model. Motivated by this, we propose two strategies to leverage the VLMs as dynamics models to enhance VLMs as world models: (i) generating synthetic trajectories by annotating large-scale key-frame pairs from videos with actions predicted by the dynamics model, then using these as weak supervision to train the world model (Section 3.2); and (ii) using the dynamics model as a verifier at test time to score candidate next observations sampled from the world model (Section 3.3).

## 3.1 FINE-TUNING CHAMELEON AS A DYNAMICS MODEL

First, we fine-tune Chameleon as a Dynamics Model (**CDM**) $p_{\text{CDM}}(a \mid o_s, o_t)$, which predicts the probability of an action given the previous and next observations. As training data, we rely on high-quality triplets from AURORA (Krojer et al., 2024) and the action recognition track of EPIC-Kitchen (Damen et al., 2018), which is based on videos with an egocentric view. We use the first and last frame in the EPIC-Kitchen video clips as the source and target observation $o_s$ and $o_t$ and the annotated action as $a$. We provide full details on CDM training data and experimental setting Appendix K. Foreshadowing the results in Section 4.2, this significantly enhances action-prediction capabilities of Chameleon by a wide margin.

## 3.2 WEAKLY SUPERVISED LEARNING FROM UNLABELLED VIDEOS

**Synthetic Trajectories.** Taking advantage of the resulting high-quality CDM, we then explore the first of our strategies to bootstrap a world model in VLMs: we annotate pairs of motion key-frames of unlabelled videos with a textual description of the action with the CDM. To ensure both scale and quality, we collect approximately 45 hours of video from Moments-in-Time (Monfort et al., 2019), Kinetics-700 (Kay et al., 2017; Carreira et al., 2019), and UCF-101 (Soomro et al., 2012), all of which consist of curated clips focused on human actions. To ensure the selected pairs of motion key-frames are meaningful, i.e., they express a valid action, we then calculate the optical flow to quantify the dynamics per frame in the video clips, and select the top-$K_f$ frames while ensuring that the interval between two selected frames is $I_f$. Specifically, we set $I_f = 20$ and $K_f = 6$ for all three datasets. This results in approximately 20K, 46K, and 21K annotated trajectory triplets from Moments-in-Time, Kinetics-700, and UCF-101, respectively. Finally, we apply a filtering strategy to further guarantee the quality of the resulting triplets. We use the CDM's predicted likelihood for each trajectory triplet $(o_s, a_{\text{CDM}}, o_t)$ as a score, and apply stratified Top-K sampling[3] to select a subset of CDM-annotated trajectory triplets. We show statistics of the scores and action classes for the selected triplets in Figure 11. We also provide one example for each dataset in Figure 3.

---

[3]The details of this algorithm are provided in Appendix G.

**Fine-tuning Chameleon as a World Model.**   Afterwards, we fine-tune Chameleon as a World Model (**CWM**), $p_{\text{CWM}}(o_t \mid a, o_s)$ on both AURORA's supervised triplets $\mathcal{D}_{\text{sup}}$ and unsupervised triples $\mathcal{D}_{\text{unsup}}$ with actions sampled from the CDM. The world model CWM is trained with maximum likelihood estimation as an objective:

$$\min_{\theta} \; \mathbb{E}_{(a,o_s,o_t)\sim\mathcal{D}_{\text{sup}}} \left[ -\log p_\theta(o_t \mid a, o_s) \right] + \mathbb{E}_{(o_s,o_t)\sim\mathcal{D}_{\text{unsup}}} \left[ \mathbb{E}_{\hat{a}\sim p_{\text{CDM}}(a|o_s,o_t)} \left[ -\log p_\theta(o_t \mid \hat{a}, o_s) \right] \right],$$
(1)

where $\theta$ are the parameters for CWM, and $\hat{a}$ is action sampled from the CDM.

**Recognition-Weighted Training Loss.**   Nevertheless, the objective in Equation 1 is limited by treating all regions of the target observation equally, even if some of them remain identical to the source whereas others change. This may result in degenerate solutions such as always copying the source. As an alternative, we therefore propose a novel training objective for world models that overcomes this assumption. This objective weights the loss of next-observation image tokens based on their importance. The intuition is that not all image patches in source and target observations contribute equally to modelling real-world transitions; instead, the model should focus on patches most indicative of the action's consequences. To this end, we leverage a recognition model $f_{\text{rec}}(w|o_s, o_t)$, which outputs token-level weights aligned with Chameleon's image token representations. These weights modulate the loss during training, emphasising learning on semantically meaningful regions and down-weighting irrelevant ones. We formulate our alternative objective as:

$$\min_{\theta} \sum_{l=1}^{L} f_{\text{rec}}(w|o_s, o_t)^{(l)} \cdot \left( -\log p_\theta(o_t^{(l)} \mid o_t^{(<l)}, o_s, a) \right),$$
(2)

where $\theta$ are the parameters of CWM and a $L$ is the number of tokens used to represent an image in Chameleon. $o_t^{(l)}$ and $o_t^{(<l)}$ represent the image tokens of $o_t$ at position $l$ and the history of previous positions, respectively. For simplicity, we use the pre-trained vector-quantised model of Chameleon as the recognition model, by computing the squared $L_2$ norm of pre-quantized features $\mathbf{z}_{o_s} \in Z_{o_s}$ and $\mathbf{z}_{o_t} \in Z_{o_t}$ where $Z_{o_s}$ and $Z_{o_t}$ are the sets of features of source and target observations, respectively. We visualise the token weights in Figure 3, which capture the effects of acting on the source observation to yield the target one.

## 3.3 TEST-TIME VERIFICATION

Finally, we introduce an inference-time strategy which harnesses the CDM as a verifier to enhance CWM performance. Inspired by recent work on scaling test-time compute (Muennighoff et al., 2025; Snell et al., 2024), we let the CWM generate $N$ candidate observations. Each candidate is paired with the source and scored by the CDM, which assigns each a predicted likelihood, interpreted as a reward. The final prediction of the CWM is selected by maximising the CDM's reward:

$$\hat{o}_t = \operatorname*{argmax}_{i\in\{1,\dots,N\}} \; p_{\text{CDM}}\left(a \mid o_s, o_t^{(i)}\right), \quad \text{where } o_t^{(i)} \sim p_{\text{CWM}}(o_t \mid o_s, a),$$

where $\hat{o}_t$ is the selected prediction.

## 4 EXPERIMENTS AND RESULTS

### 4.1 EXPERIMENTAL SETTING

**Benchmarks.**   We select AURORA-BENCH (Krojer et al., 2024) for evaluation of both dynamics and world models. This dataset provides high-quality data for action-centric edits, covering a wide array of phenomena and assessing a model's alignment with the physical world, including temporal and spatial reasoning. We choose 5 subsets: **MagicBrush** for specialised image editing, **Action-Genome (AG)** and **Something-Something (Something)** for real-world actions. **Whatsup** focuses on spatial reasoning, whereas **Kubric** contains samples from a physical engine (Greff et al., 2022).

**Baselines.**   We report Chamelon's zero-shot performance (**C-ZS**). We also fine-tune Chameleon on AURORA's training set as our first baseline (**C-FT**). We compare CWM with C-FT in both a

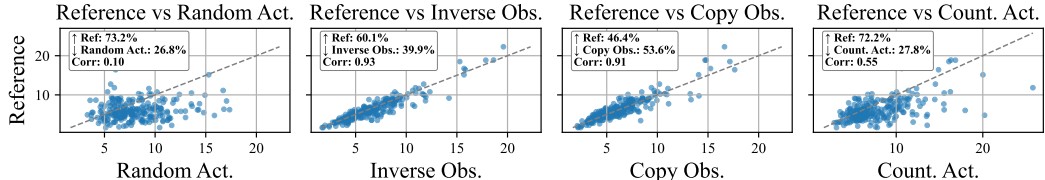

Figure 4: Comparison of negative log-likelihoods (lower values indicate stronger model preference) of the action predicted by CDM for ground-truth trajectories versus four types of negative trajectories.

single-prediction setting and in a best-of-$N$ setting. The latter provides a ceiling performance for inference-time verification with CDM. Additionally, we include three state-of-the-art diffusion models specialised for image editing as baselines, such as **PixInstruct** (Brooks et al., 2023), **GoT** (Fang et al., 2025) and **SmartEdit** (Huang et al., 2024). As a sanity check, we also report the metric scores obtained by simply copying the source observation input as the next-observation prediction (**Copy**).

**Metrics.** For next-observation prediction evaluation, following (Fang et al., 2025), we rely on **GPT4o-as-a-judge** as it is the only metric that reliably penalises Copy. In Appendix D, we show four other metrics, e.g., CLIP, which assign high scores to Copy. GPT4o-as-a-judge scores consider two criteria, one for the editing success rate and one for visual consistency with the original. We take the minimum of the two as the final score. The prompt is provided in Appendix H.

## 4.2 CHAMELEON DYNAMICS MODEL

We evaluate the dynamics models based on the textual similarity of the predicted action with the ground-truth action in AURORA-BENCH, as shown in Table 1. Our results demonstrate that fine-tuning is necessary to elicit Chameleon's ability to verbalise the dynamics from two observations. We then compare Chameleon fine-tuned on action prediction (CDM) with the fine-tuned version of another state-of-the-art VLM, VILA-U (Wu et al., 2024b). CDM is on par or superior to VILA-U fine-tuned, justifying our choice of Chameleon as a foundation model for our experiments.

Table 1: Dynamics models performance on action prediction, measured by text similarity metrics: BERTScore (BS; (Zhang et al., 2020)), ROUGE (R-1, R-L; (Lin, 2004)) and BLEU (Papineni et al., 2002).

|  | **BS** | **R-1** | **R-L** | **BLEU** |
|---|---|---|---|---|
| VILA-U SFT | 0.40 | 0.38 | 0.37 | 0.15 |
| Chameleon ZS | 0.05 | 0.09 | 0.08 | 0.00 |
| Chameleon SFT | 0.40 | 0.39 | 0.37 | 0.17 |
| Chameleon SFT + DS | **0.45** | **0.45** | **0.44** | **0.20** |

Table 1 also provides an ablation showing that downsampling trajectories from Kubric (DS) in the training data further boosts CDM performance (CDM + DS). This suggests that data sourced from simulations do not necessarily translate into better dynamics modelling in real-world examples. We use the DS version of CDM in the rest of the experiments as the best-performing dynamics model. In Figure 4, we further evaluate CDM on discriminating between ground-truth and negative trajectories, as in Section 2. Now, we observe that CDM is mostly successful in identifying manipulated trajectories as such, except for Copy. These results corroborate the feasibility of annotating actions for key-frame pairs.

## 4.3 CHAMELEON WORLD MODEL

**Automatic evaluation.** Next, we test CWM on next-observation prediction for each of the AURORA-BENCH subsets, reporting GPT4-as-a-judge scores in Figure 2. We first notice that the state-of-the-art image editing models (i.e., PixInstruct, GoT, SmartEdit) tend to specialise in the image editing benchmark, MagicBrush (5.96 and 6.71 GPT4o scores for GoT and SmartEdit). Nevertheless, in the action-centric subsets, including Action-Genome (AG), Something and Kubric, they are mostly behind CWM and even C-FT. In particular, CWM outperforms all other models in these 3 subsets, achieving gains of 18%, 4%, and 86%, respectively, over the best diffusion baselines. In addition, it boasts the highest average performance across subsets, with an 8% increase. Crucially, comparing CWM and C-FT reveals the benefit of augmenting the training data with synthetic triplets bootstrapped from the CDM, as it yields a 13% performance margin. CWM also outperforms C-FT on the best-of-$N$ setting (Amini et al., 2025), indicating the potential for inference-time verification as best-of-$N$ is effectively an oracle for its performance.

Table 2: Performance on AURORA-BENCH in terms of GPT-4o scores and two spatial-reasoning benchmarks in accuracy. For C-FT and CWM, we report the performance for both single prediction and *best-of-N*. The average scores for each model are at the bottom. We **bold** the best model overall for each subset and highlight the best and worst scores among our variants for each setting. SE: SmartEdit. We use greedy decoding for spatial reasoning evaluation, and only apply to VLMs.

| Datasets | Models | | | | | | | | |
|---|---|---|---|---|---|---|---|---|---|
| | Copy | PixInstruct | GoT | SE | C-ZS | C-FT | +*Best-of-3* | CWM | +*Best-of-3* |
| *Image-Editing* | | | | | | | | | |
| MagicBrush | 0.000 | 3.120 | 5.960 | **6.710** | 0.000 | 2.520 | 3.270 | 3.920 | 3.920 |
| AG | 0.000 | 1.200 | 1.610 | 3.080 | 0.170 | 2.480 | 2.740 | **3.640** | 3.640 |
| Something | 0.000 | 0.957 | 2.620 | 2.810 | 0.370 | 3.110 | 3.110 | 2.920 | **3.310** |
| WhatsUp | 0.000 | 0.000 | **1.580** | 0.755 | 0.146 | 0.880 | 0.980 | 0.540 | 0.540 |
| Kubric | 0.000 | 1.880 | 3.920 | 3.700 | 0.140 | 7.300 | 7.300 | 7.320 | **7.780** |
| AURORA Avg. | 0.000 | 1.430 | 3.140 | 3.410 | 0.165 | 3.260 | 3.480 | 3.670 | **3.840** |
| *Spatial Reasoning* | | | | | | | | | |
| SpatialMQA | – | – | – | – | 26.1 | 25.8 | – | **27.2** | – |
| EmbodiedSpatial-Bench | – | – | – | – | 15.1 | **21.2** | – | 17.5 | – |

**Human Evaluation.** Following (Krojer et al., 2024), we conduct a blind human evaluation comparing GoT, SmartEdit, C-FT, and our proposed CWM. We randomly sample 5 examples from each subset within AURORA-BENCH and present the outputs generated by each of the four models. Human annotators are asked to identify the best and worst generated observations based on three criteria: (1) *Realism*: the generated image should exhibit natural textures and lighting while remaining faithful to the input scene; (2) *Instruction-Following Ability*: the edit should clearly reflect the given instruction; and (3) *Over-Editing*: the modification should be minimal and focused, altering only what is necessary. Each model receives +1 point for being selected as the best, -1 for the worst, and 0 otherwise. We compute the average scores over 350 annotated samples, as reported in Table 6. The results align with automatic evaluations: image-editing models excel in the MagicBrush domain, but fall short on action-centric datasets such as Action-Genome, Something-Something, and Kubric. In contrast, CWM outperforms C-FT on all three of these datasets, highlighting its strength in next-observation prediction in real-world, action-centric trajectories.

Table 3: Detailed scores of GPT4o-as-a-judge evaluation for loss-weighting and standard training. We report the scores for **E**diting **S**uccess (**ES**) and **M**inimal **E**diting (**ME**). MB: MagicBrush, AG: Action-Genome, ST: Something-Something, WU: WhatsUp, KU: Kubric. We highlight the best and worst scores for each category.

| | Weighted | | Standard | |
|---|---|---|---|---|
| | ES (↑) | ME (↑) | ES (↑) | ME (↑) |
| MB | 3.73 | 8.17 | 3.68 | 8.46 |
| AG | 3.18 | 8.03 | 2.37 | 8.13 |
| ST | 3.32 | 7.01 | 2.78 | 7.20 |
| WU | 0.54 | 7.25 | 0.76 | 7.19 |
| KU | 7.75 | 8.49 | 7.24 | 8.70 |
| Avg. | 3.71 | 7.80 | 3.37 | 7.94 |
| GPT4o | 3.67 | | 3.58 | |

**Ablation Study on Synthetic Trajectories.** To assess the importance of extra supervision from CDM-synthetic trajectories, Table 5 reports GPT-4o's scores for this ablation. We see performance drops on most datasets—particularly on Something and AG—when the additional training data from unlabelled videos is removed, highlighting the effectiveness of bootstrapping CWM with large-scale real-world data via CDM. An exception is the WhatsUp dataset, which focuses on specific actions within a fixed scene; in this case, training in an open-domain setting may not transfer effectively.

**Ablation Study on Loss Weighting.** Based on Table 5, we also observe consistent degradation when loss weighting is removed, demonstrating the benefit of explicitly incorporating the recognition model into visual next-token prediction. To better understand the effect of loss weighting, Table 3 reports the average scores for two criteria used in the GPT-4o-as-a-judge evaluation separately: Editing Success (ES), which measures how well the model captures the intended action and performs the corresponding edit, and Minimal Editing (ME), which assesses whether the model introduces unnecessary modifications. The full distribution of GPT-4o scores is provided in Appendix I. Our

Figure 5: Ablation study of synthetic trajectories (Synth.) and loss weighting (LW) in CWM. Numbers are GPT-4o-as-judge scores (↑, average of 3 runs). MB: MagicBrush, AG: Action-Genome, ST: Something-Something, WU: WhatsUp, KU: Kubric.

Figure 6: Human evaluation results. † indicates all results whose gap with respect to CWM is significant, based on a Wilcoxon signed-rank test ($p = 0.05$). MB: MagicBrush, AG: Action-Genome, ST: Something-Something, WU: WhatsUp, KU: Kubric.

|  | **CWM** | *w/o Synth.* | *w/o LW* |
|---|---|---|---|
| **MB** | **3.48** | -0.28 | -0.22 |
| **AG** | **3.02** | -0.35 | -0.08 |
| **ST** | **3.06** | -0.18 | -0.19 |
| **WU** | **0.46** | 0.40 | 0.08 |
| **KU** | **7.14** | -0.03 | -0.33 |
| **All** | **3.43** | -0.09 | -0.15 |

|  | **GoT** | **SE** | **C-FT** | **CWM** |
|---|---|---|---|---|
| **MB** | 0.06† | **0.29**† | -0.32† | -0.03 |
| **AG** | -0.23† | -0.46† | 0.32 | **0.37** |
| **ST** | 0.00 | -0.37† | 0.18 | **0.20** |
| **WU** | **0.25** | -0.38† | 0.14 | 0.00 |
| **KU** | -0.52† | -0.22† | 0.34 | **0.40** |
| **All** | -0.09† | -0.23† | 0.13 | **0.19** |

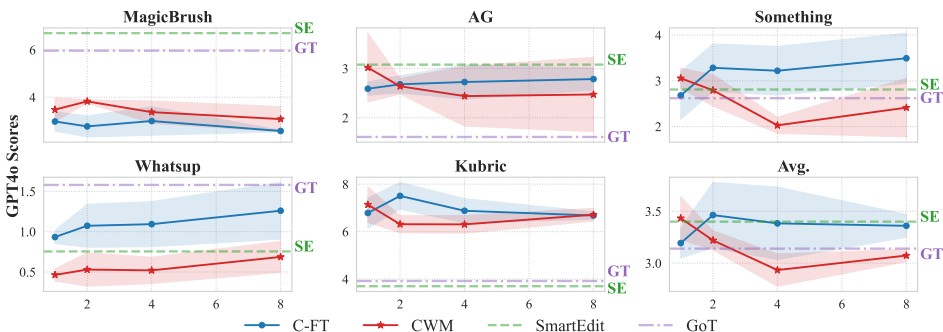

Figure 7: GPT-4o scores for test-time verification with $K$ samples, where $K \in \{1, 2, 4, 8\}$. We use a blue line for C-FT and a red line for CWM, plotting the standard deviation as the shaded area. We indicate the scores for GoT (GT) and SmartEdit (SE) as horizontal lines.

analysis reveals that the primary bottleneck for CWM remains its ability to reliably follow the instruction, as reflected by the fact that ES scores are significantly lower than ME scores. Loss weighting partly solves this problem, increasing the editing success and reducing copying behaviour, albeit at the cost of sometimes over-editing the source observation.

**Verification at Test Time.** We evaluate test-time verification using CDM in Figure 7, comparing C-FT and CWM with $K \in 1, 2, 4, 8$. Each experiment is repeated three times, and we visualise the mean and standard deviation. By increasing exploration on more candidate next observations, C-FT benefits from test-time verification on most datasets with real-world trajectories (e.g., AG, Something, WhatsUp), suggesting the effectiveness of CDM's trajectory preferences. Increasing $K$ does not always improve performance (MagicBrush, Kubric), suggesting that bootstrapping with a dynamics model that shares the same foundation model backbone may be limiting. In contrast, CWM shows no clear gain, likely because it was trained with the synthetic trajectories and has already internalised CDM's preferences—as is evident from its strong $K = 1$ performance. In summary, CDM-based verification boosts C-FT's performance to the same level as CWM, by leveraging more diverse samples at inference time rather than during training.

**Image Editing as an Auxiliary Task.** Training on the action-centric image editing task exposes the model to interactive supervision, where it predicts future observations from actions. Since AURORA contains diverse spatial relations (e.g., left/right orientation), we evaluated whether this supervision helps the model generalise beyond editing. We tested our models on two spatial reasoning benchmarks: SpatialMQA (Liu et al., 2025) and EmbodiedSpatial-Bench (Du et al., 2024). In Table 2, models trained with the world-modelling objective outperform the baseline, demonstrating that CWM transfers beyond editing and highlighting world modelling as a signal for enhancing spatial reasoning.

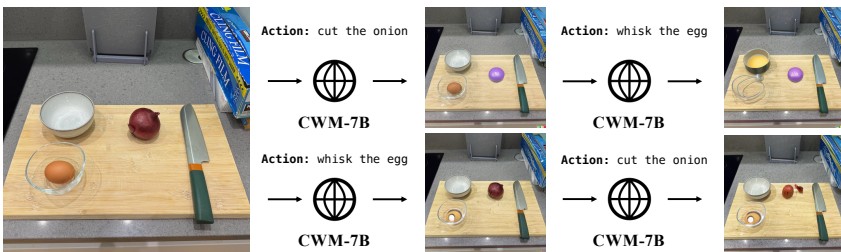

Figure 8: A qualitative case of real-world observation prediction, demonstrating CWM's ability to steer predictions using language and perform sequential predictions. More cases from AURORA-BENCH are in Appendix E.

**Qualitative Example.** Figure 8 presents a real-world example demonstrating that CWM's predicted observations can be guided through language expressing actions. CWM is also capable of iteratively generating future observations in multiple steps while maintaining consistency with previous frames.

## 5 RELATED WORK

Despite the surge in interest for world modelling (Ha and Schmidhuber, 2018; Sutton, 1988; Hafner et al., 2019b), previous works focused mostly on building specialised *ad-hoc* world models. These world models can be explicitly learnt as a visual simulator (Agarwal et al., 2025; Bruce et al., 2024; Brooks et al., 2024), or enable planning with model-based reinforcement learning (Hafner et al., 2019a; Micheli et al., 2022; Robine et al., 2023; Alonso et al., 2024; Hafner et al., 2025). Instead, we focus on leveraging large-scale multimodal foundation models (Lin et al., 2024; Chen et al., 2025; Wu et al., 2024b) to develop world models, which is more attractive due to the inductive bias they provide from their extensive training. This is possible thanks to frameworks that integrate observations, actions, and rewards into a unified sequence of tokens in autoregressive Transformers (Wu et al., 2024a), building on pioneering works such as Decision Transformers (Chen et al., 2021) and GATo (Reed et al., 2022b). Related to our work, (Chen et al., 2024) initialise the parameters of RL policies with VLMs, thus taking advantage of the abundant and general world knowledge encoded in their representations. 3D-VLA (Zhen et al., 2024) introduces a set of interaction tokens into a Large Language Model to engage with the environment as an embodied agent. (Yang et al., 2024; Soni et al., 2024) explore large-scale self-supervised learning via next token or frame prediction to build a unified model absorbing internet knowledge, learning from interaction via video.

AURORA-BENCH (Krojer et al., 2024) was the first to approach world modelling through the lens of an action-centric image editing task. With advanced native VLMs capable of the interleaved generation (Chameleon Team, 2024; Chern et al., 2024), we systematically investigate how this data may help us bootstrap a world model, implicitly stored in the VLMs, with an easier-to-train dynamics model. Most similar to our work, (Baker et al., 2022) train a dynamics model which aims to uncover the underlying action between video frames in unlabelled video frames from the Minecraft game. Through this model, they synthesise trajectories to train a policy for sequential decision making. In contrast with (Baker et al., 2022), we focus on next-observation prediction as a task to evaluate world modelling. First, this allows us to port the observation space to real-world frames, rather than simulated ones, hence assessing whether world models are well aligned with the physical environment. Second, this broadens the space of actions from a few choices to the combinatorially infinite and expressive space of language, capturing a significantly more diverse range of dynamics.

## 6 CONCLUSION

In this work, we explored whether we can develop word models from VLMs. By evaluating them on action-centric image editing AURORA-BENCH (Krojer et al., 2024), we first show that these models lack a clear preference for ground-truth real-world trajectories. To address this, we induce a dynamics model from the same VLM to bootstrap a world model using automatic annotation of unlabelled real-world videos and inference-time verification. Experiments confirm the effectiveness of both strategies, with our general-purpose world model achieving state-of-the-art performance compared to existing approaches specialised for image editing.

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

## A    LLMs Usage Declaration

We declare that the large language model (LLM) was only used to assist in minor tasks, including revising the manuscript for grammatical correctness, improving phrasing, and performing small technical implementations such as debugging code snippets. All scientific ideas, results, analyses, and conclusions presented in this paper are entirely the work of the authors.

## B    Limitations

While our approach demonstrates the effectiveness of our approaches across AURORA-BENCH, the authors would like to highlight few limitations we have discovered:

- Despite efforts to guide the model via supervised fine-tuning with loss weighting or inference-time verification (Table 2), we observe that the model may still resort to copying the source observation, especially under low sampling temperatures or ambiguous instructions.

- While we show preliminary results of language-steered observation prediction in Figure 8, fine-grained control remains limited, and understanding subtle instructions (e.g., spatial or quantitative edits) remains challenging.

- We observe variance across different runs of experiments, likely due to the sensitivity of sampling for generation in multimodal models. To address this, we report results averaged over multiple runs and include performance under the best-of-$N$ sampling distribution during inference for a robust comparison.

- We mostly conduct experiments using the native and unified VLM, Chameleon, as it is currently the only open-source VLM that supports interleaved image-text generation by default. This choice allows for fair and consistent benchmarking across our tasks. Moreover, Chameleon has demonstrated competitive performance in our settings. For example, its results are comparable to VILA-U in our dynamics prediction task. Future work should explore the generalisation to other multimodal foundation models with stronger capabilities.

## C    Broader Impact

This work develops models for action-centric image editing for visual world modelling. While our primary aim is to advance fundamental research in world modelling, we acknowledge potential risks, particularly in the generation of realistic future observations.

A core concern is the potential misuse of the models for creating deceptive visual content, including fabricated action sequences or manipulated images that imply false causality. Although the model is not explicitly designed for these tasks, its ability to generate coherent visual predictions from the linguistic action could be adapted for such uses if deployed irresponsibly.

Even in intended use, risks include over-reliance on generated outputs in downstream tasks such as robotic control, or interactive systems. Model failures—e.g., copying artefacts, hallucinations, or broken object continuity—can lead to incorrect inferences or reinforce dataset biases.

To mitigate potential misuse, we limit our model release to research purposes under a non-commercial license and clearly communicate its capabilities and limitations. We urge caution when adapting them for deployment, particularly in settings with high societal or ethical sensitivity.

## D    Model Performance on Aurora-Bench with 5 Metrics

In addition to GPT4o-as-a-judge evaluation, we further employ a diverse set of automatic metrics covering both low-level and semantic fidelity: 1) we compute the **L1 distance** between the predicted and target observation as a pixel-level metric. 2) We extract visual features and compute the cosine similarity in their respective embedding spaces for several image encoders, including (**CLIP-I** and **DINO**), to assess semantic similarity. Additionally, to measure alignment between image content and the action semantics, we compute **CLIP-T**, the similarity between the edited image and its BLIP-generated caption. These metrics are evaluated in addition to GPT4o-as-a-judge metric following

previous works in image editing (Huang et al., 2024; Fang et al., 2025; Krojer et al., 2024). We report the detailed results with 5 metrics in Table 4. We notice that copy baseline exhibits the best performance as measured by the distance-based and visual encoder-based approach, as indicated in Table 2. This poses a challenge to the reliability of the traditional metrics in fairly evaluating the action-centric image editing task. On the other hand, GPT4o-as-a-judge metric robustly assigns 0 score to Copy, indicating its robustness in detecting copying generation while putting GPT-as-a-judge as the most reliable metric to interpret.

Table 4: Model performance at MagicBrush, Action-Genome, Something, WhatsUp and Kubric on AURORA-BENCH. For C-FT and CWM We report both the model performance and their performance in the *best-of-N* distribution. We report the average GPT4o scores for each model at the bottom. We highlight the better GPT-4o scores for C-FT and CWM. We **bold** the best performance among all models, except Copy and *best-of-N* performances. SE: SmartEdit.

| Datasets | Metrics | Models | | | | | | | | |
|---|---|---|---|---|---|---|---|---|---|---|
| | | Copy | PixInstruct | GoT | SE | CM | C-FT | +*Best-of-3* | CWM | +*Best-of-3* |
| **MagicBrush** | L1 | 0.027 | 0.114 | **0.063** | 0.068 | 0.287 | 0.075 | 0.075 | 0.090 | 0.078 |
| | CLIP-I | 0.959 | 0.877 | 0.930 | **0.937** | 0.671 | 0.913 | 0.914 | 0.906 | 0.909 |
| | CLIP-T | 0.289 | 0.275 | 0.286 | 0.290 | 0.227 | 0.289 | 0.289 | **0.291** | 0.291 |
| | DINO | 0.931 | 0.761 | 0.881 | **0.894** | 0.292 | 0.883 | 0.883 | 0.864 | 0.864 |
| | GPT-4o | 0.000 | 3.120 | 5.960 | **6.710** | 0.000 | 2.520 | 3.270 | 3.920 | 3.920 |
| **AG** | L1 | 0.069 | 0.220 | 0.174 | **0.137** | 0.314 | 0.170 | 0.168 | 0.168 | 0.167 |
| | CLIP-I | 0.943 | 0.757 | 0.846 | 0.811 | 0.609 | 0.872 | 0.872 | **0.881** | 0.883 |
| | CLIP-T | 0.279 | 0.254 | 0.280 | 0.268 | 0.214 | 0.280 | 0.284 | **0.284** | 0.284 |
| | DINO | 0.929 | 0.557 | 0.785 | 0.774 | 0.258 | 0.801 | 0.817 | **0.816** | 0.816 |
| | GPT-4o | 0.000 | 1.200 | 1.610 | 3.080 | 0.170 | 2.480 | 2.740 | 3.640 | 3.640 |
| **Something** | L1 | 0.135 | 0.232 | 0.184 | **0.163** | 0.293 | 0.184 | 0.184 | 0.196 | 0.184 |
| | CLIP-I | 0.870 | 0.709 | 0.807 | 0.773 | 0.649 | **0.820** | 0.820 | 0.804 | 0.804 |
| | CLIP-T | 0.275 | 0.238 | 0.269 | 0.265 | 0.232 | **0.271** | 0.269 | 0.268 | 0.268 |
| | DINO | 0.797 | 0.453 | 0.636 | 0.662 | 0.297 | **0.675** | 0.653 | 0.666 | 0.666 |
| | GPT-4o | 0.000 | 0.957 | 2.620 | 2.810 | 0.370 | 3.110 | 3.110 | 2.920 | 3.310 |
| **WhatsUp** | L1 | 0.039 | 0.138 | 0.078 | 0.067 | 0.251 | **0.066** | 0.066 | 0.070 | 0.070 |
| | CLIP-I | 0.954 | 0.817 | **0.923** | 0.888 | 0.721 | 0.877 | 0.880 | 0.870 | 0.883 |
| | CLIP-T | 0.326 | 0.287 | **0.316** | 0.312 | 0.243 | 0.309 | 0.310 | 0.306 | 0.307 |
| | DINO | 0.908 | 0.615 | **0.850** | 0.805 | 0.424 | 0.836 | 0.841 | 0.831 | 0.838 |
| | GPT-4o | 0.000 | 0.000 | **1.580** | 0.755 | 0.146 | 0.880 | 0.980 | 0.540 | 0.540 |
| **Kubric** | L1 | 0.011 | 0.104 | **0.026** | 0.064 | 0.276 | 0.044 | 0.044 | 0.044 | 0.044 |
| | CLIP-I | 0.963 | 0.796 | 0.895 | 0.868 | 0.660 | 0.897 | 0.899 | **0.897** | 0.898 |
| | CLIP-T | 0.282 | 0.259 | 0.281 | 0.271 | 0.213 | **0.287** | 0.287 | 0.287 | 0.288 |
| | DINO | 0.955 | 0.676 | 0.857 | 0.798 | 0.161 | **0.906** | 0.906 | 0.902 | 0.902 |
| | GPT-4o | 0.000 | 1.880 | 3.920 | 3.700 | 0.140 | 7.300 | 7.300 | 7.320 | 7.780 |
| **All** | GPT-4o | 0.000 | 1.430 | 3.140 | 3.410 | 0.165 | 3.260 | 3.480 | 3.670 | 3.840 |

# E QUALITATIVE CASES

In this section, we present additional qualitative examples from AURORA-BENCH in Figure 9. We observe several common failure modes in image editing models. First, they sometimes fail to preserve the scene from the source observation (e.g., PixInstruct on Action-Genome and MagicBrush). Second, some models generate near-identical copies of the source as the target (e.g., GoT on Something-Something). Third, producing realistic outputs remains difficult, as seen in GoT's result on Kubric. Finally, maintaining object consistency is also a challenge—SmartEdit alters the object in WhatsUp, and CWM does so in Something-Something.

Despite the challenges, we also observe several positive editing behaviours from CWM. On Action-Genome, CWM correctly predicts spatial changes, such as *opening and closing a drawer*, which requires a strong understanding of the spatial concepts. In Something-Something, it is the only model to accurately capture the spatial concept of "falling down." On Kubric, it demonstrates basic counting ability by correctly adding one keyboard. In WhatsUp, CWM correctly grounds the action to the laptop, while other models mistakenly edit the monitor.

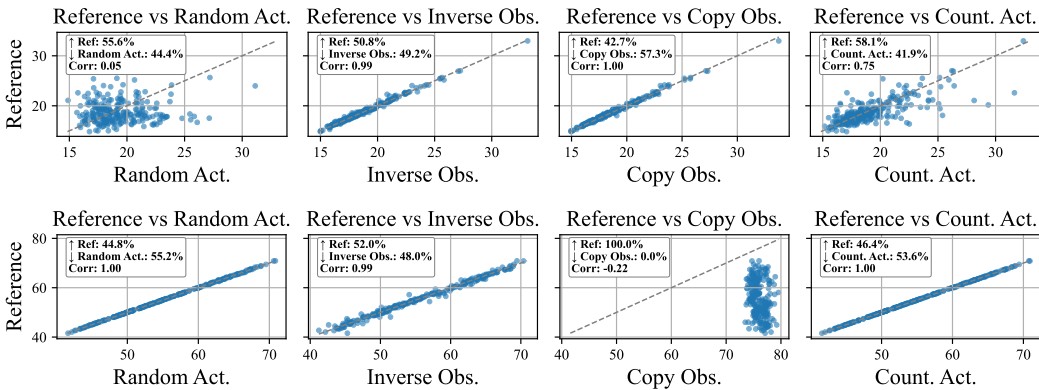

Figure 10: Comparison of predicted negative log-likelihoods (lower values indicate stronger model preference) for ground-truth real-world trajectories versus four types of negative trajectories. **Top**: Action prediction task for the dynamics model (observation × observation → action). **Bottom**: Next observation prediction task for the world model (observation × action → observation). The legend shows the percentage of times the model prefers the ground-truth trajectory (↑) over the negatives (↓).

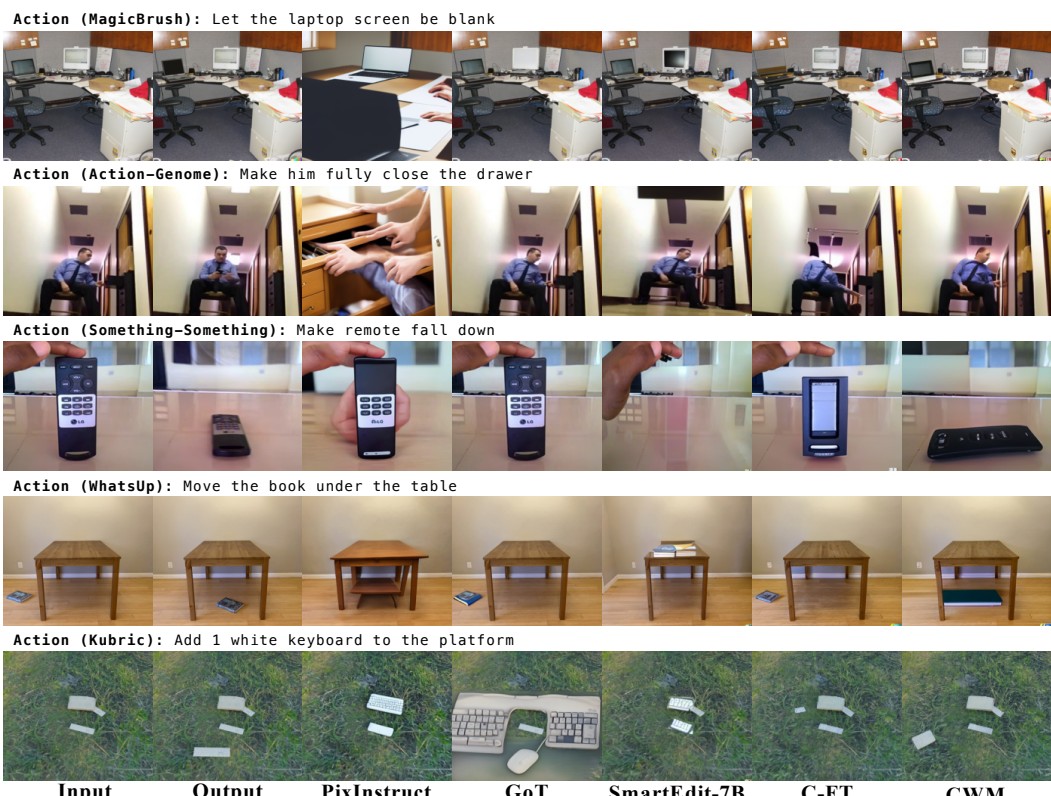

Figure 9: Qualitative examples of the predicted next observation from the state-of-the-art specialised image editing models, and our models including C-FT and CWM, on AURORA-BENCH.

## F    DETAILED DISCUSSION FOR CHAMELEON'S PREDICTED LIKELIHOODS

From Figure 10, it emerges that Chameleon-7B displays a very limited preference for the ground-truth trajectories in a zero-shot setting. In the action prediction task (top panel), there is a slightly higher tendency to favour the ground-truth; however, even in the best case (counterfactual action), the model

Table 5: Dataset statistics for the video and triplets from the trajectories annotated by CDM. **OPV**: observations (i.e., extracted key-frames) per video, **APV**: actions per video, **WPA**: words per action.

| Dataset | Video | | Triplet | | | |
|---|---|---|---|---|---|---|
| | Avg. Length | Total Length | #Samples | #Avg. OPV | #Avg. APV | #Avg. WPA |
| **MIT** | 3.04 seconds | 2.57 hours | 19,658 | 2.05 | 1.05 | 7.10 |
| **UCF-101** | 7.24 seconds | 26 hours | 10,965 | 3.00 | 2.00 | 8.96 |
| **Kinetics700** | 9.02 seconds | 18 hours | 26,959 | 2.71 | 1.71 | 7.39 |

prefers the reference in only 58.1% of the samples. The high correlation in likelihoods indicates that the VLM struggles also on visual manipulations. In the next-observation prediction task (bottom panel), the VLM mostly fails in effectively differentiating the ground truth from the negatives. An exception to this is the copy manipulation, where the model can always tell them apart. Although the underlying reason remains uncertain, one plausible explanation for this behaviour is that the model's ability to solve next-observation prediction tasks depends on their alignment with training sequences: for instance, it is plausible that Chameleon's data rarely features two identical adjacent images. In summary, Chameleon-7B does not exhibit a preference for ground-truth trajectories over negative ones, constructed through action- or observation-based manipulations.

## G  DETAILS OF PROCESSING CDM ANNOTATIONS FOR UNLABELLED VIDEOS

---

**Algorithm 1** Stratified Top-K Sampling with Action Class Uniformity

---

**Require:** Trajectory triplet set $X = \{(o_s^i, o_t^i, a^i, s^i, c^i)\}_{i=1}^N$, where $s_i$ is the predicted likelihood of $a^i$, $c_i \in \mathcal{C}$ is the class, number of samples $K$
1: Sort $X$ descending by score $s_i$
2: Initialize $S \leftarrow \emptyset$, and class_counts$[c] \leftarrow 0$ for all $c \in \mathcal{C}$
3: **while** $|S| < K$ **do**
4:     **for all** class $c \in \mathcal{C}$ in round-robin order **do**
5:         $X_c \leftarrow$ top unsampled item from class $c$ in $X$
6:         **if** $X_c \neq \emptyset$ **then**
7:             $S \leftarrow S \cup \{X_c\}$
8:             Remove $X_c$ from $X$
9:             class_counts$[c] \leftarrow$ class_counts$[c] + 1$
10:         **end if**
11:         **if** $|S| = K$ **then**
12:             **break**
13:         **end if**
14:     **end for**
15: **end while**
16: **return** $S$

---

We present the raw dataset statistics before sampling for Movements-in-Time, UCF-101 and Kinetics700 in Table 5. Figure 11 shows the distribution of CDM's predicted scores across action classes in Movements-in-Time, Kinetics700, and UCF-101. The predicted likelihoods are nearly uniform within each class, indicating that our sampling method maintains both class diversity and high overall likelihoods. The sampling procedure for CDM-annotated trajectories is detailed in Algorithm 1.

## H  PROMPT TEMPLATE FOR USING GPT4O-AS-A-JUDGE EVALUATION.

We provide the prompts used for evaluating image editing performance with GPT-4o in Figure H. We use `GPT-4o-2024-11-20`. The final score is the average of the minimum value of the two scores for each sample, as in (Fang et al., 2025).

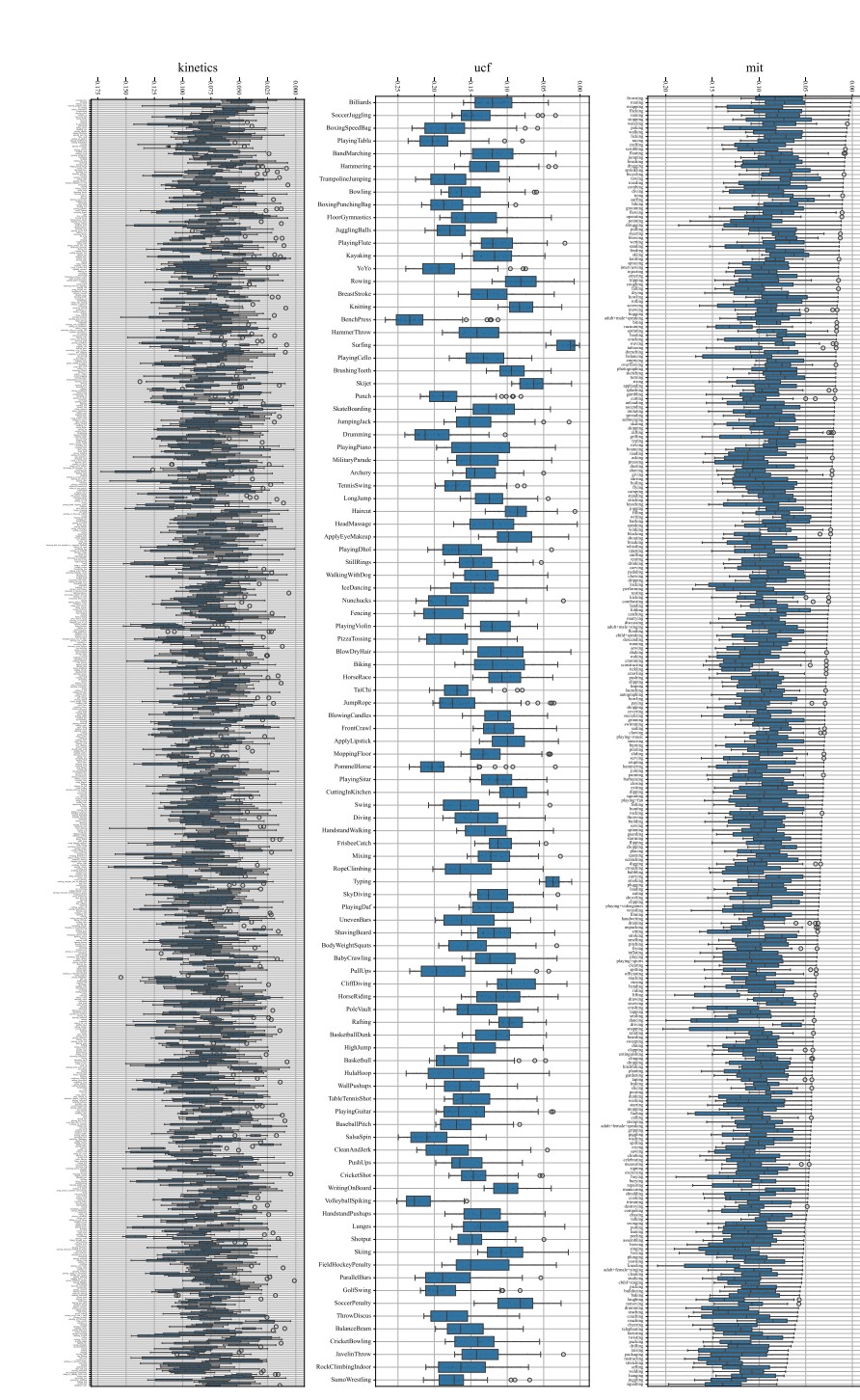

Figure 11: Distributions of triplet log-likelihoods predicted by CDM on Movements-in-Time, UCF-101, and Kinetics-700, based on 7K synthetic triplets per dataset. Triplets are uniformly sampled from each action class while maximising overall predicted likelihoods.

**Prompt Template for GPT4o-as-a-judge Evaluation**

You are a professional digital artist. You will have to evaluate the effectiveness of the AI-generated image(s) based on the given rules.

You will have to give your output in a valid way of a Python dictionary format (Keep your reasoning concise and short.):

```
{{"score":  [...], "reasoning":  "..." }}
```

and don't output anything else. Two images will be provided:

- The first being the original AI-generated image
- The second being an edited version of the first.

The objective is to evaluate how successfully the editing instruction has been executed in the second image. Note that sometimes the two images might look identical due to a failure in image editing. From a scale of 0 to 10:

- A score from 0 to 10 will be given based on the success of the editing.
  - 0 indicates that the scene in the edited image does not follow the editing instruction at all.
  - 10 indicates that the scene in the edited image follows the editing instruction text perfectly.
  - If the object in the instruction is not present in the original image at all, the score will be 0.
- A second score from 0 to 10 will rate the degree of minimal editing in the second image.
  - 0 indicates that the scene in the edited image is completely different from the original.
  - 10 indicates that the edited image can be recognised as a minimally edited yet effective version of the original.

Put the score in a list such that: output score = [score1, score2], where score1 evaluates the editing success and score2 evaluates the degree of the minimal editing.
Editing instruction: {instruction}

# I  DETAILED GPT4O SCORES FOR EDITING SUCCESS AND MINIMAL EDITING

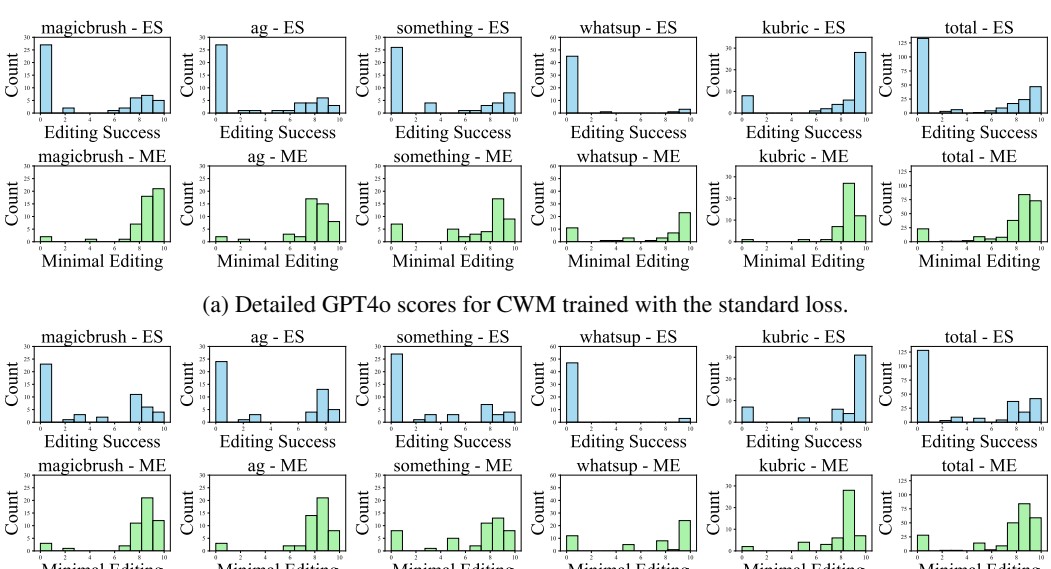

(a) Detailed GPT4o scores for CWM trained with the standard loss.

(b) Detailed GPT4o scores for CWM trained with the $L_2$-weighted loss.

Figure 12: GPT4o scores' distributions of editing success (ES) and minimal editing (OE) for CWM trained with standard loss or our loss-weighting method.

Figure 12 shows the distribution of editing success (ES) and minimal editing (ME) scores for standard training and loss-weighted training. Loss weighting tends to improve editing success, with a modest trade-off in minimal editing quality in most of the datasets.

# J  IMPLEMENTATION DETAILS

# K  CHAMELEON DYNAMICS MODEL

We fine-tune the Chameleon-7B checkpoint from the Anole-7B version (Chern et al., 2024) to predict the action given a pair of observations, framed as an action-prediction task. The model is trained on a merged dataset from Action-Genome, Kubric, MagicBrush, Something-Something from AURORA's annotated trajectories, and 15K EPIC-Kitchens processed by us. We downsample Kubric's trajectories to 10K. Training is performed for 10 epochs with a batch size of 64, using a learning rate of 2e-4 and cosine scheduling (500 warm-up steps). We use bfloat16 mixed-precision training and apply LoRA (Hu et al., 2022) for parameter-efficient fine-tuning (rank 16, $\alpha = 32$, dropout 0.05). Only the completion loss is used to optimise the generation of action. Training is conducted on 4 NVIDIA-H100-80GB-HBM3 GPUs using DeepSpeed for distributed optimisation.

# L  C-FT BASELINE

We fine-tune the Chameleon-7B checkpoint from the Anole-7B version (Chern et al., 2024). The model is trained on a combined dataset from Action-Genome, Kubric, MagicBrush, and Something-Something, formatted as the image editing task. We downsample Kubric's trajectories to 10K. Training is conducted for 40 epochs with a batch size of 96 using the AdamW optimiser and a cosine learning rate scheduler (learning rate of 5e-4, 400 warm-up steps). We use mixed-precision training with bfloat16 and apply LoRA (Hu et al., 2022) for efficient fine-tuning (rank 16, $\alpha = 32$, dropout 0.05). We only train the model with the truncated loss from the completion part. We use 4 NVIDIA-H100-80GB-HBM3 GPUs with DeepSpeed for distributed training. During inference, we apply a logits processor to mask out non-image tokens, set the temperature to 1, and use top-1

sampling. We observe that temperature is critical in controlling model behaviour: lower values often cause the model to copy the source observation instead of generating meaningful edits.

## L.1 CHAMELEON WORLD MODEL

We fine-tune the Chameleon-7B checkpoint from the Anole-7B version (Chern et al., 2024). The model is trained on a combined dataset from Action-Genome, Kubric, MagicBrush, Something-Something from AURORA's annotated trajectories, together with 7K trajectories from Movements-in-Time, 7K trajectories from UCF-101 and 7K trajectories from Kinetics700, formatted as the image editing task. Again, we downsample Kubric's trajectories to 10K. Training is conducted for 40 epochs with a batch size of 96 using the AdamW optimiser and a cosine learning rate scheduler (learning rate of 5e-4, 400 warm-up steps). We use mixed-precision training with bfloat16 and apply LoRA (Hu et al., 2022) for efficient fine-tuning (rank 16, $\alpha = 32$, dropout 0.05). We only train the model with the truncated loss from the completion part, but we weight the image tokens using $L_2$ strategy as introduced in Section 3. We use 4 NVIDIA-H100-80GB-HBM3 GPUs with DeepSpeed for distributed training. We use the same hyperparameters as C-FT during the inference time.

## L.2 COMPUTING RESOURCES

All training experiments were conducted on a compute node equipped with 4× NVIDIA H100 80GB GPUs, 256 CPU cores, and 256GB of memory. The total GPU hours required for training C-FT, CWM, and CDM were approximately 200, 400, and 100 hours, respectively.

For inference, we used a single NVIDIA A100 80GB GPU with 8 CPU cores and 128GB memory. Inference for C-FT and CWM takes approximately 1 GPU hour per model. When applying verification with $K = 8$, inference time increases to around 8 GPU hours. CDM only takes around 0.3 GPU hours for inference.

## L.3 ASSETS AND LICENSES

In this section, we list the public assets we used in this paper and the corresponding links.

**Datasets.** We include the detailed license and URL for the datasets we used in this paper.

- AURORA and AURORA-BENCH (Krojer et al., 2024): MIT license, the reader can find the corresponding version we use in this paper in `https://github.com/McGill-NLP/AURORA`.
- Movements-in-Time (Monfort et al., 2019): BSD-2-Clause license and its own License for Non-Commercial Use, the reader can find the corresponding version we use in this paper in `http://moments.csail.mit.edu/`.
- UCF-101 (Soomro et al., 2012): unknown license, the reader can find the corresponding version we use in this paper in `https://huggingface.co/datasets/flwrlabs/ucf101`.
- Kinetics700 (Kay et al., 2017; Carreira et al., 2019): Creative Commons Attribution 4.0 International License, the reader can find the corresponding version we use in this paper in `https://research.google/pubs/the-kinetics-human-action-video-dataset/`.
- EPIC-Kitchens (Damen et al., 2018): Creative Commons Attribution-NonCommercial 4.0 International License, the reader can find the corresponding version we use in this paper in `https://epic-kitchens.github.io/`.

**Implementation.** We use the other following code for the implementations:

- Transformers (Wolf et al., 2020): Apache-2.0 license. We use the 4.47.0 version, following the link at `https://github.com/huggingface/transformers`.
- DeepSpeed: We use the 0.14.4 version, following the link at `https://github.com/deepspeedai/DeepSpeed`.

**Model.** We use the following models or checkpoints for the implementations:

- Chameleon (Chameleon Team, 2024): Chameleon Research License, the reader can find the corresponding version we use in this paper in `https://github.com/facebookresearch/chameleon`.
- Anole-7B (Chern et al., 2024): Chameleon Research License and MIT License, the reader can find the corresponding version we use in this paper in `https://github.com/GAIR-NLP/anole`.
- VILA-U (Chern et al., 2024): MIT License, the reader can find the corresponding version we use in this paper in `https://github.com/mit-han-lab/vila-u`.
- SmartEdit (Huang et al., 2024): Apache-2.0, the reader can find the corresponding version we use in this paper in `https://huggingface.co/TencentARC/SmartEdit-7B`.
- GoT (Fang et al., 2025): MIT License, the reader can find the corresponding version we use in this paper in `https://github.com/rongyaofang/GoT`.
- PixInstruct (Brooks et al., 2023): PixInstruct customised license, the reader can find the corresponding version we use in this paper in `https://github.com/timothybrooks/instruct-pix2pix`.

## M  DETAILS OF HUMAN EVALUATION

We conducted a human evaluation using a custom-built interface, with the full interface and instructions shown in Figure 13. A total of 14 participants were recruited, all of whom are PhD-level graduate students or higher. Participation was voluntary. Each participant was asked to evaluate 25 samples, which typically required 15–20 minutes to complete.

The evaluation process, including recruitment, instructions, and data processing and storage, followed our institution's ethical guidelines for human subject research. All participants were informed of the purpose of the study and provided consent. No personally identifiable information was collected, and all data were stored and analysed in accordance with privacy standards.

## N  SAFEGUARDS

CWM performs observation prediction through image generation and, while its outputs are task-specific, we acknowledge that any generative model may carry potential for misuse. To mitigate these risks, we commit to the following safeguards upon release:

The model will be released solely for research purposes under a license that prohibits commercial use or any other harmful applications. The GitHub repository will include clear usage guidelines and terms of use, aligned with responsible AI principles.

We will include a disclaimer that the model is intended only for academic research in controlled environments. The datasets used for training are publicly available, action-centric image editing benchmarks that do not include sensitive or personally identifiable content.

Given the targeted nature of our model and the safeguards in place, we believe the risk of misuse is limited. Nonetheless, we encourage responsible use and welcome feedback from the community regarding potential improvements to safety.

Figure 13: The screenshot for the instructions given to participants and the interface developed for conducting the evaluation.

