# OpenReview forum: "Bootstrapping World Models from Dynamics Models in Multimodal Foundation Models"
_ICLR.cc/2026/Conference — ICLR 2026 Conference Withdrawn Submission_

### Official Review · Reviewer_5B7o · 2025-10-30

**Soundness:** 2
**Presentation:** 2
**Contribution:** 2
**Rating:** 4
**Confidence:** 4

**Summary:**

This paper investigates the extent to which Vision-Language Models (VLMs) possess implicit world models (observation, action → next_observation) and dynamics models (observation, next_observation → action). The authors first empirically demonstrate that existing open-source VLMs, such as Chameleon-7B, struggle to distinguish ground-truth trajectories from manipulated ones, indicating a weak inherent world model. A key finding is that fine-tuning a VLM to act as a dynamics model is significantly easier than training it directly as a world model.

Based on this insight, the paper proposes two main strategies to "bootstrap" a world model using a pre-trained dynamics model:

1. Weakly Supervised Learning: A fine-tuned dynamics model (CDM) is used to annotate large-scale, unlabelled video datasets (e.g., Kinetics, UCF-101) with actions, creating a large corpus of synthetic (obs, action, obs') triplets. This synthetic data is then used to fine-tune the VLM into a world model (CWM). The authors also introduce a recognition-weighted loss to focus training on image regions that change.

2. Inference-Time Verification: The dynamics model is used as a verifier to score and rank multiple candidate next_observation samples generated by the world model at test time, selecting the one with the highest likelihood.

Experiments on the AURORA-BENCH benchmark show that the proposed CWM, trained with synthetic data, achieves performance competitive with or superior to state-of-the-art specialized image editing models, particularly on real-world action-centric subsets. The inference-time verification is also shown to be an effective training-free alternative.

**Strengths:**

1. Novel and Insightful Core Idea: The central concept of bootstrapping a world model from a dynamics model is clever and well-motivated. The empirical finding that learning p(action | obs, obs') is easier than p(obs' | obs, action) provides a strong foundation for the entire work.

2. Thorough and Rigorous Experimentation: The paper features a comprehensive evaluation suite. It starts by clearly demonstrating the problem, uses a relevant benchmark (AURORA-BENCH), compares against strong baselines, and employs both automatic (GPT-4o) and human evaluation. The justification for the choice of metric is sound.

3. Effective Methodologies: Both proposed bootstrapping strategies—weakly supervised learning with synthetic data and inference-time verification—are shown to be effective. The ablation studies clearly demonstrate the positive impact of both the synthetic data and the proposed recognition-weighted loss.

4. Significant Problem Domain: The work addresses the crucial challenge of building world models, which is fundamental for progress in embodied AI, planning, and physical reasoning. By grounding this in powerful foundation models, the paper connects two important areas of research.

**Weaknesses:**

1. Limited Generality: The most significant weakness is that all experiments are based on the Chameleon-7B model. While the authors provide a reason for this choice, it makes it difficult to assess whether the core findings—especially the relative ease of learning dynamics models—are a general property of VLMs or specific to the Chameleon architecture. The claims would be much stronger if validated on at least one other V-L architecture, even if it required a different implementation approach.

2. Analysis of Computational Cost: The inference-time verification method is shown to be effective, but its computational cost is substantial (e.g., increasing inference time by a factor of K, where K can be 8 or more). The paper notes the increased GPU hours but does not provide a clear cost-benefit analysis. For instance, how does the performance of C-FT + Best-of-8 compare to CWM (single prediction) in terms of total computation (training + inference)? A discussion on the trade-offs between training-time data augmentation and test-time compute would strengthen the paper.

3. Ambiguity in "Easier to Learn" Claim: The paper claims learning a dynamics model is "substantially easier," based on empirical results. This is a key finding, but the paper could benefit from a deeper discussion on the potential reasons why. Is it because the output space (a short text string for an action) is less complex and lower-dimensional than the output space for an image? Is it because predicting an action from two states is a more constrained classification/regression-like task than the highly generative task of predicting a future state? A more nuanced discussion would add depth to this central claim.

4. Mixed Performance Gains: While the CWM model performs very well on action-centric datasets like AG and Something, it is still outperformed by specialized models on the image editing-focused MagicBrush dataset. The average performance gains reported in Table 2 are positive but modest. This suggests that while the approach is promising, it is not a universal solution and has its own set of trade-offs, which could be discussed more explicitly.

**Questions:**

1. Regarding generalizability: Could the authors comment on the potential to apply their bootstrapping principle to other types of multimodal models, such as those based on diffusion architectures? Do you foresee any fundamental architectural barriers that would prevent a dynamics model from effectively guiding or training a diffusion-based world model?

2. Regarding the "easier to learn" hypothesis: Could you elaborate on your intuition for why action prediction (obs, obs' -> act) is substantially easier for a VLM to learn than next-observation prediction (obs, act -> obs')? Does this primarily relate to the complexity of the output space (text vs. image tokens) or to the nature of the underlying reasoning required?

3. Regarding inference cost: Could you provide a more direct comparison of the computational cost (e.g., in total GPU-hours) required to reach a certain performance level? For example, what is the cost of training CWM versus the cost of using C-FT with test-time verification (e.g., K=8) to achieve a similar GPT-4o score on the AG subset? This would help readers better understand the practical trade-offs.

4. Regarding the test-time verification results in Figure 7: CWM shows little to no improvement from verification, which the paper attributes to it having already "internalised CDM’s preferences." Could this also be interpreted as a limitation? If the generator (CWM) and verifier (CDM) are trained from the same base model and similar data distributions, they might share the same blind spots, limiting the effectiveness of the verification process. Have you considered using a verifier from a completely different model family to see if it could provide more significant gains?

---

> ### Comment · Reviewer_5B7o · 2025-11-26
> **A few thoughts following up on other reviews**
>
> I've been thinking through the comments from the other reviewers (especially Reviewer Zx9X's points on the presentation confusion and Reviewer 69ER's concerns about novelty).
>
> While I still find the core idea of bootstrapping world models (WM) from dynamics models intuitively appealing (as I mentioned in my original review), I have to admit that the concerns raised by others regarding the WM vs. simple image editing are making me reconsider my initial assessment.
>
> I hope the authors can engage with these points in the rebuttal. Specifically, beyond the "easier to learn" claim I asked about, I'm keen to see if you can address the distinction between learning a true underlying physics/world representation versus just overfitting to the editing task. If the author can give a strong explanation, then I am willing to improve my score.

---

### Official Review · Reviewer_69ER · 2025-10-30

**Soundness:** 2
**Presentation:** 3
**Contribution:** 2
**Rating:** 2
**Confidence:** 3

**Summary:**

This paper presents an action centric image editing method which relies on fine-tuning a pretrained VLM capable of image generation (Anole, derived from Chameleon). This approach, framed as learning a world model, first fine-tunes the VLM to predict actions from a pair of frames. This is used to expand the aurora training set, by annotating pairs of video frames with a corresponding action. The VLM is then fine-tuned again on aurora and the annotated data to predict the edited frame from an action instruction. The model is evaluated on aurora-bench, against other image editing methods, showing promising performance.

**Strengths:**

The paper is, for the most part, well written and easy to follow. Framing the task of image editing as learning a world model is an interesting point of view, which can potentially inspire to investigate image editing from a new angle.

The proposed approach is sensible, and well ablated, although it would habe been nice to get detailed results from the human study. Building a larger dataset to achieve better performance makes sense, and leads to performance improvements. Results are promising compared to state of the art editing methods.

**Weaknesses:**

My main concern for this work is the lack of novelty. While framing the task as learning a world model is original, the proposed method has very little innovation, and is essentially a sequence of fine-tuning tasks on pre-existing datasets and models with fixed architectures. Building a large training set by relying on VLM annotations is often the starting point of many multi-modal image generation or editing papers, and often requires more complex work than simple fine-tuning on a pre-existing dataset [1].

The weighted loss is interesting, but not grounded in literature, as alternative approaches to token-based image editing are not discussed. This is partially due to the world model framing, which redirects the related work towards world models and action centric models, failing to review other editing methods and their limitations. Image editing is a very active field of research, many approaches [1] rely on VLM to build datasets and edit images. While these approaches are different, it is important to highlight how similar concepts are leveraged in the literature. Even editing methods being compared to are not discussed in the related works.

Lastly, training a dynamics models seems wasteful with little impact. It could be replaced with pre-trained model VILA-U (which has very similar performance to CDM), and the verification step at inference time has little to no impact on the fine-tuned model. As authors admit in the limitation section in the appendix, using the same base model to verify itself can be suboptimal. It would have been interesting to see if using VILA-U as a verifier would have had a different impact.

Regarding the evaluation, the high performance on the Kubric dataset appears to skew the average in favour of the proposed approach, with more balanced results for other datasets. Results mainly highlight that this approach is designed for action-centric editing, while others were designed for the magic brush dataset.

The world model framing is interesting, but seems to be an overstatement given the scenario considered. There is no evidence that the VLM has learned a representation of the world, or gained an understanding of the dynamics involved in transitioning from image A to its edited version. For example, a model that has learned a “make the remote fall” edit might not be able to do “show the remote during the fall”.

[1] omniedit, ICLR 2025; omnigen, arxiv 2025; fireedit, CVPR 2025

**Questions:**

-	Overall, presenting image editing as learning a world model is interesting, and it is true that models with a latent representation of the world would be able to achieve high quality action-centric edits. However, it is unlikely that the proposed methodology achieves this. Besides achieving edits better, is there any evidence that the model has learned a latent representation of the world?
-	The world model framing prevents this work from being properly grounded in the image editing literature. I would recommend for authors to highlight how this approach differs from pre-existing work, not just from the action-centric point of view, but also from a methodological point of view. Why is this approach better and innovative?
-	Are all models compared to trained on the aurora dataset? It would be great to make this clear to ensure all differences in performance are due to methodological choices and not training data

---

### Official Review · Reviewer_Zx9X · 2025-10-31

**Soundness:** 2
**Presentation:** 1
**Contribution:** 2
**Rating:** 2
**Confidence:** 2

**Summary:**

The authors explore to what extent open-source VLMs can be finetuned to perform as world models and inverse dynamics models. They first conduct an experiment to check if VLMs can discriminate between ground truth real-world trajectories and manipulated trajectories. The analysis is used as the rationale to finetune the Chameleon 7B model to become an inverse dynamics model, called CDM, since it showed that VLMs find it harder to perform out-of-the-box world modelling, as opposed to inverse action prediction. The CDM model is then used to bootstrap a world model, resulting in the CWM (Chameleon World Model). The authors perform both automatic (GPT4o-as-judge) and human evaluation on the CWM model and present 2 ablation studies to assess the impact of synthetic trajectories and loss weighting. The evaluation is conducted on an image editing task on Aurora-bench for both models and as benchmarks they choose 3 SOTA image editing models: PixInstruct, GoT and SmartEdit and zero-shot Chameleon. The authors conclude that CWM improves real-world subset performance by 15% over SOTA (judged by GPT-4o) and achieves the highest average human evaluation across all subsets.

**Strengths:**

- The authors address an interesting question, understanding if we can finetune VLMs to act as world and action prediction models. However, the results are not very clearly presented and I found it difficult to follow the details of the experiments conducted. I recommend this work goes through a bit more refining in presentation and be resubmitted again.
- Fair variety of benchmarks to test the models on for the image editing task.
- Good ablation studies to assist the mechanics introduced (synthetic trajectories and loss weighting)
- Authors express willingness to opensource code and models.

**Weaknesses:**

- Authors refer to the action prediction model as a dynamics model, but that is incorrect. These are referred to as inverse dynamics models $ p(a_t \mid o_t, o_{t+1}) $, where you predict the action that caused a transition between two states or observations. Dynamic models (or forward dynamics models) predict the next state or observation given the current state and an action $ p(s_{t+1} \mid s_t, a_t) $
- The paper is really difficult to follow due to a large number of presentation issues highlighted below, I think the work needs to be more polished before it's ready for publication. The model names keep on changing between main body and figures and tables, making it incredibly difficult to follow through and takes away from the paper's main experiments and results.

**Questions:**

Suggestions and questions:
- Figure 1 should be referenced in the main body
- In Section 2, from line 138, it reads like 1 type manipulates the action (Random Action) and the rest manipulate observations, but the first sentence indicates differently (2 manipulate the action and 2 the observations).
- Line 148 misses table reference, should it reference Figure 2?
- Line 153 - should this be the "copy observation" task (as opposed to "inverse observation") where Owen 2.5VL-TB is associated with 67.34%?
- Line 161 - footnote - there is a typo on 1) => "they are publicly accesible"
- Should Figure 3 be updated to include all the datasets represented in the figure (missing AG, Kinetics700 and MIT)?
- Names in Table 21 should match names on line 269 (e.g. C-FT for Chameleon SFT, which is also referred to as CDM) and then we have Chameleon-SFT+DS which is in the main body referred to as CDM+DS. This is quite confusing for readers to follow
- On line 314 (section 4.3), the reference should be for Table 2, instead of Figure 2
- It's unclear what the last line in the **Avg. GPT4o** row in Table 3 stands for - it would be good to clarify in the table description or main body.
- Figure 5 should be Table 5, as referenced on line 372
- Similarly, Figure 6 should be Table 6, as on line 359 - additionally, given Table 6 is discussed before, it would be good for 5 and 6 to be swapped to follow the order in the main paper body.

---

### Official Review · Reviewer_CujL · 2025-11-01

**Soundness:** 3
**Presentation:** 4
**Contribution:** 2
**Rating:** 6
**Confidence:** 4

**Summary:**

This paper shows that today’s vision–language models (like Chameleon-7B) aren’t very good “world models”. They don’t reliably prefer real action to result image sequences over fake ones. But they can be fine-tuned much more easily to do the inverse-dynamics task (given before/after images, say what action happened). The authors exploit that: first they train a strong dynamics model, then use it in two ways to bootstrap a better world model: (1) label lots of unlabeled video key-frame pairs with actions to create extra (obs, action, next-obs) training data, and (2) at inference, generate several candidate edited images and let the dynamics model score which one is most consistent. With these two strategies plus a loss that focuses the model on the image patches that actually changed, their “Chameleon World Model” beats or matches state-of-the-art image-editing baselines on the action-centric AURORA-BENCH (especially real-world subsets) according to GPT-4o and human evals.

**Strengths:**

1. The paper notices a real asymmetry — inverse dynamics (obsₜ, obsₜ₊₁ → action) is easier to teach than full world modelling (obsₜ, action → obsₜ₊₁).
2. The paper show (i) weakly supervised training with synthetic triplets and (ii) inference-time verification/reranking.
3. The framework beats or matches specialised image-editing diffusion baselines on the action-centric subsets and show gains with both GPT-4o-as-judge and human evals.

**Weaknesses:**

1. Almost everything is done on Chameleon
2. Even though the paper talks like “world modeling,” almost all experiments are one-step (or tiny multi-step) image prediction, not long rollouts where compounding error actually bites.

**Questions:**

Why do authors specifically pick Chameleon? There are also many video-language foundations models that might better model state transitions or sequential image generation.

---

### Note · Authors · 2026-01-03

I have read and agree with the venue's withdrawal policy on behalf of myself and my co-authors.